# Development and Validation of the Acute PNeumonia Early Assessment Score for Safely Discharging Low-Risk SARS-CoV-2-Infected Patients from the Emergency Department

**DOI:** 10.3390/jcm11030881

**Published:** 2022-02-08

**Authors:** Sergio Venturini, Elisa Pontoni, Rossella Carnelos, Domenico Arcidiacono, Silvia Da Ros, Laura De Santi, Daniele Orso, Francesco Cugini, Sara Fossati, Astrid Callegari, Walter Mancini, Maurizio Tonizzo, Alessandro Grembiale, Massimo Crapis, GianLuca Colussi

**Affiliations:** 1Department of Infectious Diseases, ASFO “Santa Maria degli Angeli” Hospital of Pordenone, 33170 Pordenone, Italy; sergio.venturini@asfo.sanita.fvg.it (S.V.); sara.fossati@asfo.sanita.fvg.it (S.F.); astrid.callegari@asfo.sanita.fvg.it (A.C.); massimo.crapis@asfo.sanita.fvg.it (M.C.); 2Department of Emergency Medicine, ASFO “Santa Maria degli Angeli” Hospital of Pordenone, 33170 Pordenone, Italy; elidoc@libero.it (E.P.); nicarcidiacono@gmail.com (D.A.); silvia.daros@asfo.sanita.fvg.it (S.D.R.); laura.desanti@asfo.sanita.fvg.it (L.D.S.); 3Division of Internal Medicine and Emergency Medicine Residency Program, Department of Medicine, 1st Floor, Building n.8, Piazzale Santa Maria della Misericordia 1, 33100 Udine, Italy; rossella.carnelos@gmail.com; 4Department of Anesthesia and Intensive Care, ASUFC “Santa Maria della Misericordia” University Hospital of Udine, 33100 Udine, Italy; sd7782.do@gmail.com; 5Department of Emergency Medicine, ASUFC Hospital of San Daniele, 33038 San Daniele del Friuli, Italy; francesco.cugini@gmail.com; 6Department of Nephrology, ASFO “Santa Maria degli Angeli” Hospital of Pordenone, 33170 Pordenone, Italy; walter.mancini@asfo.sanita.fvg.it; 7Department of Medicine, ASFO “Santa Maria degli Angeli” Hospital of Pordenone, 33170 Pordenone, Italy; maurizio.tonizzo@asfo.sanita.fvg.it (M.T.); alessandro.grembiale@asfo.sanita.fvg.it (A.G.)

**Keywords:** prognostic score, COVID-19, decision curve, prediction model, NEWS2, observational study

## Abstract

A continuous demand for assistance and an overcrowded emergency department (ED) require early and safe discharge of low-risk severe acute respiratory syndrome coronavirus 2 (SARS-CoV-2)-infected patients. We developed (*n* = 128) and validated (*n* = 330) the acute PNeumonia early assessment (aPNea) score in a tertiary hospital and preliminarily tested the score on an external secondary hospital (*n* = 97). The score’s performance was compared to that of the National Early Warning Score 2 (NEWS2). The composite outcome of either death or oral intubation within 30 days from admission occurred in 101 and 28 patients in the two hospitals, respectively. The area under the receiver operating characteristic (AUROC) curve of the aPNea model was 0.86 (95% confidence interval (CI), 0.78–0.93) and 0.79 (95% CI, 0.73–0.89) for the development and validation cohorts, respectively. The aPNea score discriminated low-risk patients better than NEWS2 at a 10% outcome probability, corresponding to five cut-off points and one cut-off point, respectively. aPNea’s cut-off reduced the number of unnecessary hospitalizations without missing outcomes by 27% (95% CI, 9–41) in the validation cohort. NEWS2 was not significant. In the external cohort, aPNea’s cut-off had 93% sensitivity (95% CI, 83–102) and a 94% negative predictive value (95% CI, 87–102). In conclusion, the aPNea score appears to be appropriate for discharging low-risk SARS-CoV-2-infected patients from the ED.

## 1. Background

An insidious type of pneumonia caused by a new coronavirus infection appeared at the end of 2019 in Wuhan, China, and soon became a pandemic disease. At the end of 2020, Italy had 2,038,759 cumulative cases of infection and 71,620 deaths, accounting for 2.6% and 4.1% of the global burden of infections and deaths at that moment, respectively [1]. The causal agent, severe acute respiratory syndrome coronavirus 2 (SARS-CoV-2), induces a severe form of interstitial pneumonia (COVID-19) in a low percentage of subjects [2]. However, because of its high infectivity, the absolute number of patients with COVID-19 needing hospital admission and intensive care support increased rapidly. The rapid overload of medical wards and intensive care units (ICUs) with SARS-CoV-2-infected patients quickly saturated hospitals’ capacity, with important consequences for the global survival of the Italian population [3].

The emergency department (ED) has been the first point-of-care at which patients with respiratory symptoms and suspicion of SARS-CoV-2 infection present for assistance. Patients with COVID-19 can be seriously ill at presentation and need immediate hospitalization or can be paucisymptomatic but at risk of complications that would need hospital care. Discrimination of the low-risk patient to discharge early and safely at home is fundamental for reducing ED overcrowding. This discrimination should be performed by objective criteria based on clinical conditions, vital signs, and other significant prognostic factors that should help the clinician determine the probability of sudden adverse events or serious clinical progression.

Several prognostic scores have been adopted in EDs to classify patients at admission and assign them to the most appropriate level of assistance (triage). The National Early Warning Score 2 (NEWS2) is one of the most popular prognostic scores. It was developed in the United Kingdom (UK) [4] and has been widely applied in EDs in Europe to classify patients at risk of serious clinical progression [5]. In the UK, NEWS2 is recommended for the risk stratification of patients with COVID-19, but NEWS2’s performance was questioned because of its low ability to predict ICU admission and death [6].

In this study, we propose the acute PNeumonia early assessment (aPNea) score for the early detection of adverse outcomes in SARS-CoV-2-infected patients. This prognostic score was developed and validated during the second Italian wave of SARS-CoV-2 infections in the ED of a tertiary hub hospital and preliminarily tested in a small external cohort of a secondary spoke hospital. We present the results of the aPNea score’s performance and metrics compared with those of the NEWS2 calculated on the same cohorts.

## 2. Patients and Methods

### 2.1. Study Design

This study was designed as a retrospective–prospective observational cohort study. We included consecutive patients who presented to the ED of the hub hospital of Pordenone and the spoke hospital of Spilimbergo, between October 2020 and November 2020, with a documented SARS-CoV-2 infection. The hospitals serve an area of about 450,000 inhabitants in the northeast of Italy. We included data on patients of all sexes, with an age equal to or greater than 18 years and a nasopharyngeal swab positive for SARS-CoV-2 infection. Patients with unclear or undefined genomic test results and pregnant women were excluded. In all patients, we collected information on general clinical characteristics, vital signs, the Glasgow Coma Scale (GCS), general laboratory tests, arterial blood gas analysis, results of the six-minute walk test, and results of lung imaging. General laboratory tests included blood urea nitrogen (BUN), plasma creatinine, C-reactive protein (CRP), procalcitonin, and D-dimer levels. Plasma D-dimer concentration was measured in ng/mL of fibrinogen equivalent units (FEU). SARS-CoV-2 infection was assessed by genomic analysis of viral RNA on a nasopharyngeal swab at presentation to the ED. The viral RNA was analyzed by a reverse transcriptase-polymerase chain reaction (RT-PCR) and COVID-19 was diagnosed according to the World Health Organization (WHO)’s guidance [7]. The result of the genomic test was available within 6 h after submission to the laboratory service. Based on a first medical evaluation, patients were discharged, kept in the observation unit, or hospitalized according to vital signs or disease severity by a subjective physician determination. NEWS2 and aPNea scores were calculated with data on patients collected within 24 h after ED admission. The outcome of the study was the combination of either oral intubation or death within 30 days, whichever occurred first. All-cause mortality and oral intubation were assessed by checking patients’ electronic records after ED admission. Although rehospitalization or ED readmission could have been better outcomes for assessing safe discharge, 30-day death and oral intubation were outcomes easily accessible in our database. This is because, during the pandemic emergency, not all discharged patients were managed in the same hospital and some patients received home assistance. Therefore, we considered this composite outcome the most appropriate for deciding which patient to discharge safely.

This study was approved by the Institutional Review Board of the University of Udine (protocol number 083-2021, approved on 16 November 2021). Written informed consent to using personal clinical data was collected from all participants or family members when the participant was critically ill or deceased. Data were anonymized before storage and analysis. This study adheres to the “Transparent Reporting of a multivariable prediction model for Individual Prognosis Or Diagnosis” (TRIPOD) statement [8].

### 2.2. Score Calculation

The NEWS2 score was calculated according to the original definition by evaluating respiratory rate, arterial oxygen saturation, need for oxygen support, systolic blood pressure, heart rate, consciousness by the Acute, Confusion, Voice, Pain, Unresponsive (ACVPU) scale, and body temperature [4]. The aPNea score was built in an arbitrary manner according to parallel clinical experiences from other groups [9]. Specifically, we assigned a maximum of 5 points to SARS-CoV-2-positive patients that needed hospitalization for any reason independent of COVID-19 severity or patients with signs of severe respiratory failure. We graded respiratory failure based on the Horowitz index and the alveolar to arterial oxygen (A-a O_2_) gradient by assigning 4 points to less severe respiratory dysfunction and 2 points to altered A-a O_2_ gradient or silent hypoxia diagnosed after a positive 6-min walk test. The importance of lung imaging for predicting active COVID-19 was translated to 1 or 2 points assigned to the involvement of a progressive lung disease. Finally, we assigned 1 point to factors that have been associated with COVID-19 activity, such as a body temperature greater than 38 °C and an elevated plasma D-dimer level (Table 1).

The Horowitz index was calculated as the arterial oxygen partial pressure in mm Hg divided by the fraction of inspired oxygen [10]. The A-a O_2_ gradient was measured by the equation [11]:$$713\times fraction\;of\;inspired\;O_{2}-\frac{partial\;pressure\;of\;CO_{2}}{0.8}-arterial\;partial\;pressure\;of\;O_{2}$$

The A-a O_2_ gradient expected by age was estimated by the equation [11]:$$\left(Patient^{\prime }s\;age+10\right)/4$$

### 2.3. Six-Minute Walk Test

SARS-CoV-2-infected patients with respiratory symptoms but a baseline pulse oxygen saturation higher than 94% by finger pulse oximeter measurement performed a six-minute walk test. Briefly, the patient walked for six minutes along a circuit of 30 m in length and pulse oxygen saturation and heart rate were measured with a finger pulse oximeter at baseline and after every minute up to the end of the test. Subjective respiratory symptoms and the total distance walked were recorded according to the standard method [12]. The test was interrupted when physical exhaustion occurred or when the pulse oxygen saturation fell below 80%. A pulse oxygen saturation below 90% during the walk test or a total distance walked of less than 400 m was considered a positive test.

### 2.4. Lung Imaging

All SARS-CoV-2-infected patients underwent first-level lung imaging by ultrasound or plain radiography within 24 h of ED admission. Images suggestive of interstitial pneumonia or highly suspected patients independent of first-level lung imaging usually were confirmed by a computerized tomography (CT) scan with contrast enhancement. Thoracic CT scans were evaluated by a radiologist who expressed the percentage of lung involved as recommended in the diagnostic work-up of suspected COVID-19 patients [13]. Lung CT scan, ultrasound imaging, or plain radiography (whichever one in the order was available) was classified by the examiner as high-risk according to lung involvement equal to or greater than 50% [14].

### 2.5. Statistical Methods

Normally distributed continuous variables are presented as the mean ± standard deviation. Non-normally distributed variables are presented as the median and interquartile range (IQR). Normal distribution was assessed by looking at the histogram of the variable distribution and by the Shapiro–Wilk test. The difference between means was analyzed with Student’s t-test. The non-parametric Mann–Whitney U test was used to compare skewed distributed variables. The difference in proportions was analyzed by the Chi-squared test. Test statistics involving multiple comparisons were corrected with the Bonferroni method. The aPNea score was developed retrospectively on a dataset of SARS-CoV-2-infected patients seen during the first 30 days of the study period. Then, the aPNea score was validated prospectively on a subsequent cohort of infected patients admitted to the same ED. In addition, the aPNea score was tested retrospectively on a small external cohort of SARS-CoV-2-infected patients from another hospital.

To determine the predictive probability of scores for the composite outcome, aPNea and NEWS2 scores were included in two univariate logistic regression models. For each model, we calculated and plotted parameters that described discrimination and calibration performances [15]. As a measure of discrimination, we calculated the area under the receiver operating characteristic (AUROC) curve and its 95% confidence interval (CI) for predicting the composite outcome. AUROCs of scores were compared with the DeLong’s test. The Brier score is the mean square error between predicted probabilities and observed proportions of the composite outcome [16]. It is a composite measure of discrimination and calibration for a binary model. The lower the Brier score, the better the calibration model, with 0 indicating a perfect model [16]. The Brier score of each score was compared between score models by Wald’s test. The Spiegelhalter z-score test assesses the calibration part of the Brier score against a perfectly calibrated model; if significant, then miscalibration is present [16].

Calibration was visualized by plotting the polynomial function of predicted probabilities versus observed proportions of the composite outcome with its 95% confidence band according to the method proposed by Finazzi et al. [17] The *p*-value of the test statistic evaluates the tendency of the model to diverge from the equality line (bisector) along the increasing predicted probability. The range of probabilities where the model overestimates (under the bisector) or underestimates (over the bisector) the outcome occurrence was evidenced in the plot [17].

The cost of discharging (opt-out) a patient for a given predicted outcome probability $\left(p_{t}\right)$ was based on the true negative rate discounted by the false negative rate (opt-out net benefit) represented by the following equation [18,19]:$$opt\_out\;net\;benefit=\frac{true\;negatives}{total\;sample\;size}-\frac{false\;negatives}{total\;sample\;size}\times \frac{p_{t}}{1-p_{t}}$$

The outcome probability threshold for avoiding hospitalization was evaluated by the analysis of decision curves performed according to Vickers et al. [18]. The net benefit standardized by the number of true negative patients represents the percent reduction in unnecessary hospitalizations without missing outcomes that is obtained by using the prediction model. The standardized opt-out net benefit with the 95% CI of each score model at different outcome probabilities was compared with a “hospitalize-all” strategy of zero net benefit [19]. The score cut-off value corresponding to the established better outcome probability for discharging patients in the development cohort was used to categorize patients as “low-risk” in the validation and external cohorts. Sensitivity, specificity, and both positive and negative predictive values were calculated with established cut-off for each score model in the validation and external cohorts. The false omission rate is the proportion of false negative patients incorrectly discharged and is defined as the complement of the negative predictive value. The false discovery rate is the proportion of false positive patients incorrectly hospitalized and is defined as the complement of the positive predictive value. The sensitivity and specificity between different scores were compared by McNemar’s test. Positive and negative predictive values were compared by a generalized score statistic.

We considered significant a test statistic with a *p*-value of <0.050. All statistical analyses were performed with the free R software version 4.0.3 [20].

## 3. Results

In 2020, the hub hospital admitted 1980 patients with SARS-CoV-2 infection. After applying the exclusion criteria, we included data from 458 infected patients. All patients presented to the ED because of mild to severe respiratory symptoms and concerns about SARS-CoV-2 infection or for underlying illness or other medical reasons and tested positive for SARS-CoV-2 infection at hospital screening. The main reasons for hospital admission independent of COVID-19 severity were chest pain with ECG abnormalities, dehydration because of gastro-intestinal loss, hemodynamic instability, and incidental trauma. The composite outcome occurred in 101 patients and the proportion of the composite outcome increased over the increased ranges of both scores (Figure 1). The aPNea score model was developed with data on the first 128 patients and validated with data on the subsequent 330 patients in the hub hospital. The score was also tested on 97 patients of the external spoke hospital. In the same cohorts, we calculated the NEWS2 score.

Regarding patients without the composite outcome, patients with the outcome were older, had a higher prevalence of obesity, diabetes, and hypertension, and had a history of congestive heart failure, coronary artery disease, and cerebrovascular disease. These patients had lower GCS and arterial oxygen saturation and a higher body temperature, heart rate, and respiratory rate. They had also higher plasma creatinine, BUN, CRP, procalcitonin, D-dimer, aPNea, and NEWS2 scores and a lower estimated glomerular filtration rate (Table 2). Patients of the validation cohort had a higher CRP level, body temperature, calculated A-a O_2_ gradient, aPNea score, and NEWS2 score than patients in the development cohort. In addition, they had lower arterial oxygen saturation, a lower arterial oxygen partial pressure, and a lower Horowitz index. Patients of the external cohort were older and had a higher prevalence of hypertension, higher BUN and CRP levels, a higher body temperature, higher calculated and expected A-a O_2_ gradients, lower arterial oxygen saturation, and a lower Horowitz index than patients in the development cohort (Table 3).

### Score Performance and Validation

In the development cohort, the AUROC of the aPNea score model was higher than that of NEWS2, whereas in the validation cohort, they were equivalent (Table 4). In the validation cohort, both the aPNea and NEWS2 score models showed no miscalibration (Figure 2 and Table 4), and between scores there was no difference in the overall calibration performance (Table 4).

In Figure 3, we report the decision curve analysis that shows the standardized opt-out net benefit of choosing a strategy based on score models instead of “hospitalize-all” in both the development and validation cohorts. Looking at the 95% CI of the development cohort, the NEWS2 model was not better than a “hospitalize-all” strategy in the low-risk probability range, whereas the aPNea model showed a significant net benefit from the low-risk threshold of 5%. Because we were looking for a safe discharge tool, we considered an outcome risk probability threshold of 10%. At this risk threshold, in the development cohort, the aPNea score showed a reduction in the number of unnecessary hospitalizations by 34% (95% CI, 12–64) without missing outcomes. Conversely, the NEWS2 score was not significant (reduced hospitalizations by 0%, 95% CI from −25% to 37%). The outcome probability of 10% in the development cohort corresponded to an aPNea score of 5 and a NEWS2 score of 1. By looking at the decision curve analysis in the validation cohort with a 10% outcome probability threshold, the aPNea score reduced unnecessary hospitalizations by 27% (95% CI, 9–41). NEWS2 again was not significant (reduced hospitalizations by 11%, 95% CI from -3% to 23%, Figure 3).

We calculated score metrics for the aPNea score with a cut-off of 5 and a NEWS2 cut-off of 1 in both the validation and external cohorts (Table 5). In both cohorts, we observed elevated sensitivity and a negative predictive value of the aPNea score, and metrics of the aPNea score were superior to those of NEWS2 in the external cohort. In the validation cohort, 36% of patients were discharged with a false omission rate of 4.2% and 64% were hospitalized with a false discovery rate of 67%. By applying the aPNea cut-off to the external cohort, 36% of patients would have been discharged with a false omission rate of 5.7% and 64% hospitalized with a false discovery rate of 58%.

## 4. Discussion

The WHO suggested using predictive scores for identifying patients with COVID-19 to hospitalize [21]. However, no agreement on a dedicated prognostic score has been reached among experts. The PRIEST study was the largest study that assessed the performance of pre-existing prognostic scores for the occurrence of death or organ support in about 21,000 suspected COVID-19 patients from 70 EDs across the United Kingdom (UK) [22]. This study compared several models: the WHO decision-making algorithm for acute respiratory infection, the pneumonia-severity-related score CURB-65 and the simplified form CRB-65, the Pandemic Modified Early Warning Score (PMEWS), Swine Flu Adult Hospital Pathway (SFAHP), and NEWS2. The best performance was shown by the NEWS2 score (AUROC = 0.77) and the worst by the WHO algorithm (AUROC = 0.61). All scores showed a high sensitivity (≥95%) at the expense of a low specificity (≤30%) [22]. Unfortunately, the PRIEST study did not suggest any cut-off to apply decision-making strategies to COVID-19 patients. Comparing these results to those of our study, we confirmed the moderate discrimination performance of the NEWS2 model in suspected COVID-19 patients, but we note the non-inferior performance and promising results of our aPNea score.

Other smaller studies that have compared predictive models for COVID-19-related mortality in EDs show variable performances of the NEWS2 score with AUROCs ranging from 0.67 to 0.81 [23,24,25]. Bradley et al. compared NEWS2, CURB-65, and the quick Sepsis-Related Organ Failure Assessment (qSOFA) score in 830 patients at risk of COVID-19 from seven hospitals. The authors concluded that all the prognostic scores underestimate the risk of death in COVID-19 patients and showed a low capacity of the scores to identify low-risk patients at the usual thresholds [25]. In addition, the authors observed that when the variables composing each score were individually analyzed, independent predictors of mortality were those variables related to the respiratory system (respiratory rate and fraction of inspired oxygen) compared with those related to circulation [25]. In our study, the NEWS2 and aPNea scores had equivalent discrimination and overall calibration performances in predicting the composite outcome, but the aPNea score better discriminated low-risk patients than the NEWS2 score. According to the observation of Bradley et al., regarding NEWS2, the aPNea score includes more variables about the involvement of the respiratory system, to which have been given the highest weight for calculating the score value. This could have produced the finer classification of low-risk patients by the aPNea score compared with NEWS2. We should say that NEWS2 remains simpler to calculate than the aPNea score. However, with more detailed information, easily available in all EDs, we can improve our triage process for SARS-CoV-2-infected patients.

Another important advantage of the aPNea score was the inclusion of clinical conditions that need hospital admission beyond COVID-19 severity. These conditions represent important risk factors for hospital admission and mortality in COVID-19 patients [26]. The mean number of diseases associated with death in a previous Italian cohort of COVID-19 patients was 2.7 ± 1.6, and only 0.8% of dead patients had no pre-existing clinical conditions [2]. Hypertension, diabetes, obesity, and respiratory and cardiovascular diseases are the most common comorbidities that predicted in-hospital mortality in COVID-19 patients [27,28]. In our study, regarding patients without the outcome, those with the outcome were most often hypertensive, diabetic, or obese and had a higher prevalence of pre-existing cardiovascular disease. Therefore, the results of our study using the aPNea score cut-off to discharge patients safely from the ED indicate that patients without serious associated diseases and with a low comorbidity burden should be discharged.

Inflammation and vascular thrombosis have been recognized as important pathophysiological mechanisms of chronic diseases and have been associated with increased cardiovascular risk and severe complications in COVID-19 patients [29]. In this study, patients with the outcome presented greater levels of CRP and D-dimer, which are markers of inflammation and pro-thrombotic activation, respectively [29]. The concomitance of advanced age, chronic diseases, inflammation, and risk of vascular thrombosis, stressed by the SARS-CoV-2 infection, justifies a large proportion of COVID-19-related mortality [29,30]. These observations were reflected in the development of the aPNea score because we included plasma D-dimer levels as a specific marker of the thrombo-inflammatory risk associated with COVID-19. In fact, D-dimer is among the markers most frequently associated with COVID-19 complications and mortality in previous studies [31,32].

Recently, several prognostic models for COVID-19 mortality have been developed, but only one, the 4C Mortality score, has received attention after a meta-analysis of 107 models [33]. The 4C Mortality score for predicting in-hospital mortality was developed and validated on more than 35,000 COVID-19 patients in 260 UK hospitals. This score comprises eight variables (age, sex, comorbidities, respiratory rate, arterial oxygen saturation, GCS, urea, and CRP), ranges from 0 to 21 points, and by the cut-off of 3 showed an AUROC of 0.77 (95% CI, 0.76–0.77), a Brier score of 0.171, and a negative predictive value of 99%. By this cut-off, the 4C Mortality score classified as low-risk 7.4% of patients with a false omission rate of 1.2%. In addition, the same authors showed that the 4C Mortality score was superior to 15 other pre-existing scores that predict COVID-19-related mortality with an AUROC that ranges from 0.61 to 0.76 [34]. Although we cannot perform a direct comparison between scores, the discrimination and calibration performances of our aPNea score appear to be similar to those of the 4C Mortality score. In addition, our preliminary results suggest that the aPNea score has the advantage of identifying a larger proportion of true negative patients. Therefore, this score could become a promising candidate for discharging SARS-CoV-2-infected patients early and safely from the ED.

This study has some limits to discuss. First, the assignment of points for building the aPNea score was arbitrary and not based on weighting of the relative risk of prognostic factors, as expected. However, during the first pandemic wave of SARS-CoV-2 infection, we did not have any information on the relative risk of specific factors for COVD-19 hospitalization. Bella et al. in Sassari showed that among the strong clinical predictors of a COVID-19 diagnosis are fever and respiratory failure. In addition, the absence of lung involvement at CT scan had a very strong negative predictive value [9]. Bertazzoni et al. in Rome confirmed the high value of a CT scan and the importance of the Horowitz index for COVID-19 diagnosis [35]. Therefore, the aPNea score was built by considering parallel clinical experiences of other Italian EDs through the reports of the Italian Society of Emergency Medicine [9,35] and information available in the international literature on the emergent role of lung imaging [13,36], silent hypoxia [37,38], and D-dimer [32]. Second, ours was a miniscule multicenter study. The aPNea score was developed and validated on two consecutive cohorts from a tertiary hospital and preliminarily tested on a small cohort from an external secondary hospital. However, both hospitals appertain to the same regional area. This approach limits the interpretation and extrapolation of our results. Because our findings are related solely to an isolated experience of COVID-19 management, application of the aPNea score to other EDs with different patient characteristics should be verified. Although our score model showed promising results, we should perform a larger prospective multicenter study. Third, we developed the score retrospectively and performed validation prospectively over a short period during the second Italian COVID-19 wave. This decision was contingent on the need to reduce rapidly ED overcrowding during the increasingly pressing COVID-19 pandemic. Therefore, samples for the development and validation dataset were not randomly selected from the entire cohort, as commonly should be done in model development. This procedure determined different cohort characteristics, in which the validation and external cohorts contained more severe cases of COVID-19 than the development cohort did. This difference could have influenced the development of the prediction model and could have required a finer degree of model tuning. However, we observed that the aPNea model performed sufficiently well to maintain its good discrimination and calibration performances in those cohorts with more severe cases. These findings suggest the excellent adaptability of the aPNea score to the varied clinical conditions potentially observed in other EDs. Fourth, the aPNea score was developed as a decision-making tool for discharging SARS-CoV-2-infected patients early and safely from the ED. Therefore, the score does not identify high-risk patients to hospitalize. This is an important point, since the high rate of false positive patients hospitalized after applying the aPNea cut-off remained a significant management problem. Other tools for improving classification of and properly addressing these high-risk patients after ED admission are needed. A potential solution could be the implementation of new score models or combining different prognostic models, as we have previously suggested for the management of sepsis in the ED of a secondary hospital [39]. Finally, NEWS2 predicts poor prognosis and it may be inappropriate for driving the discharge rather than the hospitalization of patients. However, a low NEWS2 score is associated with a low risk of clinical deterioration, and we have shown previously that NEWS2 has the best negative predictive value for adverse outcomes when compared with other prognostic scores in septic patients [39]. In addition, NEWS2 has been widely adopted in Emergency Departments and has shown the best prognostic value in assessing COVID-19 severity in several studies [40]. Therefore, we have compared our prognostic score to the best validated score for COVID-19 severity available at the time.

In summary, the aPNea score has good discrimination and calibration performances that are not inferior to those of the more frequently adopted NEWS2. The score shows excellent adaptability to different COVID-19 severities and is superior to NEWS2 in discriminating low-risk patients suitable for early and safe discharge from the ED.

## Figures and Tables

**Figure 1 jcm-11-00881-f001:**
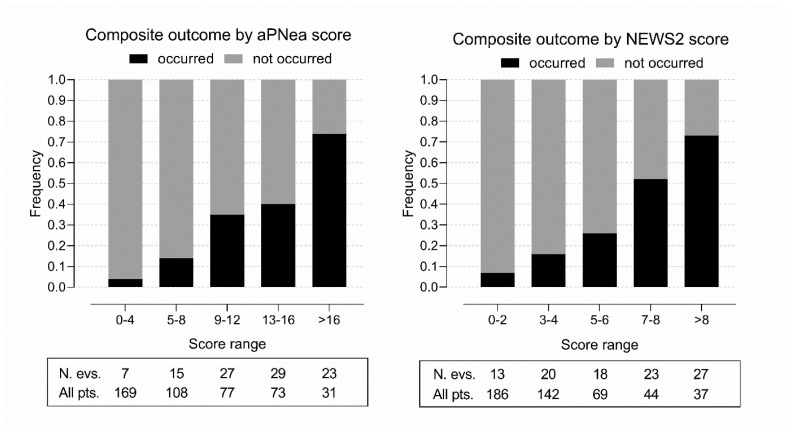
Bar graph presenting the frequency of the composite outcome by the aPNea and NEWS2 score ranges in the hub hospital. Below (rectangle) are reported the number of events (N. evs.) and the number of patients (All pts.) in each score range.

**Figure 2 jcm-11-00881-f002:**
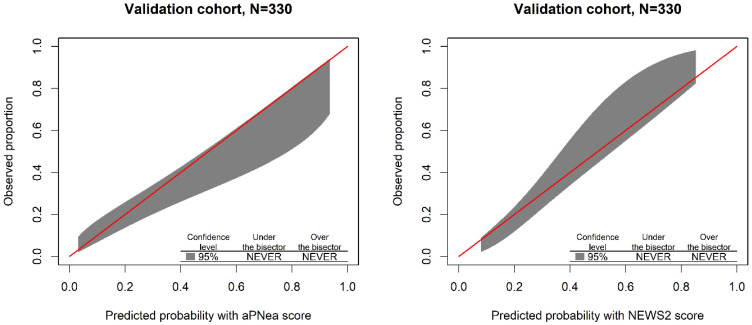
Calibration graph presenting the polynomial function and 95% confidence band of the relationship between the predicted outcome probability and the observed proportions in the validation cohort for each score model. The red bisector represents the equality line, where the predicted probability and observed proportion correspond. The range of probabilities where the model overestimates (under the bisector) or underestimates (over the bisector) the outcome is evidenced in the plot. If the model does not overestimate or underestimate the outcome outside the 95% confidence band, the word NEVER is used.

**Figure 3 jcm-11-00881-f003:**
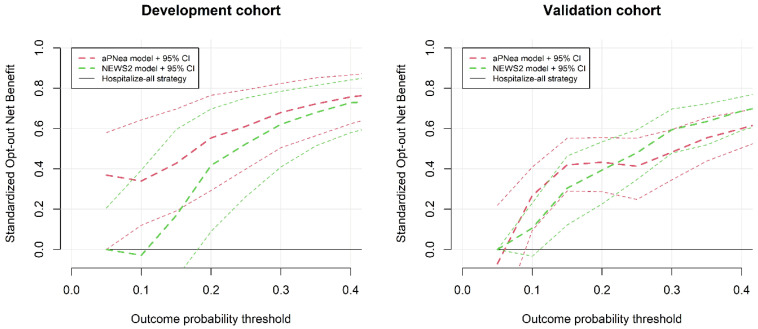
Decision curves for the development and validation cohorts presenting the standardized opt-out net benefit with the 95% confidence interval by increasing the outcome probability threshold for each score model compared to a “hospitalize-all” strategy of zero net benefit.

**Table 1 jcm-11-00881-t001:** Acute PNeumonia early assessment (aPNea) score points.

	Points
Need for hospitalization independent of COVID-19 severity	5
Respiratory rate >30 acts per minute	5
Arterial partial pressure of oxygen <60 mm Hg	5
Horowitz index <300	5
Horowitz index <330 and measured A-a O_2_ gradient ≥2 times than expected by age measured A-a O_2_ gradient <2 times than expected by age	4 2
Positive 6-min walk test	2
Lung parenchymal involvement ≥50% on the best available lung imaging	2
Lung parenchymal involvement <50% on the best available lung imaging	1
Fever >38 °C in the previous 7 days	1
Plasma D-dimer levels >1000 ng/mL FEU	1

The Horowitz index was calculated as the arterial partial pressure of oxygen in mm Hg divided by the fraction of inspired oxygen. A-a, alveolar to arterial; FEU, fibrinogen equivalent units.

**Table 2 jcm-11-00881-t002:** Variables summary and comparison according to the composite outcome occurrence after follow-up at 30 days in the entire cohort of patients from the hub hospital.

Variable	All Patients	Deceased or a Need for Oral Intubation	Alive without a Need for Oral Intubation	*p*
Sample number	458	101	357	-
General clinical characteristics				
Age (years)	67 ± 17	78 ± 13	63 ± 17	<0.001
Female sex (*n* (%))	189 (41)	36 (36)	153 (43)	0.209
Obesity (*n* (%))	99 (22)	30 (30)	69 (19)	0.029
Diabetes (*n* (%))	61 (13)	20 (20)	41 (11)	0.045
Hypertension (*n* (%))	239 (52)	76 (75)	163 (46)	<0.001
Past or active smoking (*n* (%))	64 (14)	15 (15)	49 (14)	0.747
Congestive heart failure (*n* (%))	32 (7.0)	16 (16)	16 (4.5)	<0.001
Coronary artery disease (*n* (%))	50 (11)	18 (18)	32 (9.0)	0.018
Stroke or TIA (*n* (%))	39 (8.5)	18 (18)	21 (5.9)	<0.001
30-day death (*n* (%))	87 (19)	87 (19)	-	-
Oral intubation (*n* (%))	28 (6.1)	28 (28)	-	-
Vital signs				
Glasgow Coma Scale	14.8 ± 1.0	14.4 ± 2.0	15.0 ± 0.3	<0.001
Body temperature (°C)	37.0 ± 1.0	37.2 ± 1.0	36.9 ± 0.9	0.022
SBP (mm Hg)	136 ± 22	135 ± 28	136 ± 20	0.770
Mean arterial pressure (mm Hg)	98 ± 14	95 ± 19	98 ± 12	0.130
Heart rate (beats per min)	84 ± 16	91 ± 21	82 ± 14	<0.001
Respiratory rate (acts per min)	20 ± 6	25 ± 8	19 ± 4	<0.001
Arterial oxygen saturation (%)	94 ± 5	90 ± 7	95 ± 4	<0.001
Biochemical variables				
Plasma creatinine (mg/dl)	0.92 (0.75–1.16)	1.19 (0.86–1.60)	0.88 (0.74–1.08)	<0.001
BUN (mg/dl)	17 (13–25)	28 (19–47)	15 (12–21)	<0.001
eGFR (ml/min/1.73 m^2^)	75 (57–91)	53 (41–78)	78 (64–93)	<0.001
C-reactive protein (mg/dl)	4.6 (1.4–10.7)	12 (5.9–17.8)	3.2 (0.9–7.5)	<0.001
Procalcitonin (ng/mL)	0.03 (0.01–0.11)	0.19 (0.06–0.50)	0.02 (0.01–0.07)	<0.001
D-dimer (ng/mL FEU)	705 (439–1306)	1038 (654–1606)	619 (399–1158)	<0.001
Prognostic scores				
aPNea score	6 (1–11)	12 (8–14)	5 (1–8)	<0.001
NEWS2 score	2 (1–4)	5 (3–8)	2 (1–3)	<0.001

TIA, transitory ischemic attack; SBP, systolic blood pressure; BUN, blood urea nitrogen; eGFR, estimated glomerular filtration rate; FEU, fibrinogen equivalent units.

**Table 3 jcm-11-00881-t003:** Comparison of validation and external cohort variables with those of the development cohort.

Variable	Development	Validation	External
Sample number	128	330	97
Outcome prevalence (*n* (%))	25 (20)	76 (23)	28 (29)
General clinical characteristics and laboratory variables			
Age (years)	65 ± 19	67 ± 16	71 ± 18 *
Female sex (*n* (%))	55 (43)	134 (41)	30 (31)
Obesity (*n* (%))	19 (15)	80 (24)	25 (26)
Diabetes (*n* (%))	15 (12)	46 (14)	21 (22)
Hypertension (*n* (%))	56 (44)	183 (55)	65 (67) **
Past or active smoking (*n* (%))	12 (9.4)	52 (16)	15 (15)
Congestive heart failure (*n* (%))	12 (9.4)	20 (6.1)	9 (9.3)
Coronary artery disease (*n* (%))	15 (12)	35 (11)	9 (9.3)
Stroke or TIA (*n* (%))	10 (7.8)	29 (8.8)	13 (13)
30-day death (*n* (%))	23 (18)	64 (19)	27 (28)
Oral intubation (*n* (%)])	5 (4.0)	23 (7.0)	3 (3.1)
Plasma creatinine (mg/dl)	0.9 (0.7–1.1)	0.9 (0.8–1.2)	1.0 (0.8–1.3)
Blood urea nitrogen (mg/dl)	16 (11–23)	18 (13–26)	21 (16–28) ***
eGFR (ml/min/1.73 m^2^)	78 ± 28	73 ± 29	69 ± 27
C-reactive protein (mg/dl)	3.1 (0.8–7.0)	5.1 (1.7–12.1) ***	5.1 (1.6–11.1)*
Procalcitonin (ng/mL)	0.02 (0.01–0.08)	0.04 (0.01–0.13)	0.05 (0.01–0.14)
Vital signs			
Glasgow Coma Scale	14.9 ± 0.6	14.8 ± 1.1	14.7 ± 1.1
Body temperature (°C)	36.8 ± 0.8	37.1 ± 1.0 **	37.3 ± 0.9 ***
Systolic blood pressure (mm Hg)	132 ± 19	137 ± 22	135 ± 19
Mean arterial pressure (mm Hg)	96 ± 13	98 ± 14	96 ± 11
Heart rate (beats per minute)	82 ± 15	85 ± 17	83 ± 14
Respiratory rate (acts per minute)	20 ± 6	20 ± 6	20 ± 6
Arterial oxygen saturation (%)	96 (94–98)	95 (91–97) ***	95 (93–97) *
Variables used for calculating the aPNea score			
Hospitalization COVID-independent (*n* (%))	23 (18)	39 (12)	21 (22)
Arterial O_2_ partial pressure (mm Hg)	75 (64–84)	67 (59–78) ***	68 (61–80) *
Horowitz index	332 ± 86	290 ± 96 ***	279 ± 97 ***
Positive walk test (*n* (%))	16 (13)	31 (9.4)	1 (1.0) **
Calculated A-a O_2_ gradient (mm Hg)	40 (28–55)	47 (33–93) **	53 (39–150) ***
Expected A-a O_2_ gradient (mm Hg)	21 (17–24)	22 (18–24)	24 (19–25) *
Lung involvement ≥ 50% on X-ray (%)	32 (25)	113 (35)	39 (42)
Lung involvement ≥ 50% on US (%)	32 (25)	113 (35)	39 (42)
Lung involvement ≥ 50% on CT (%)	32 (25)	113 (35)	39 (42)
D-dimer (ng/mL FEU)	656 (423–1155)	709 (453–1356)	814 (559–1569)
Prognostic scores			
aPNea score	4 (1–8)	7 (2–12)*	6 (2–11)
NEWS2 score	1 (1–3)	3 (1–4)*	2 (1–4)

Bonferroni-corrected * *p* < 0.050; ** *p* < 0.010; *** *p* < 0.001 vs. the development cohort; TIA, transitory ischemic attack; SBP, systolic blood pressure; eGFR, estimated glomerular filtration rate; US, ultrasound; CT, computerized tomography; FEU, fibrinogen equivalent units.

**Table 4 jcm-11-00881-t004:** aPNea score and NEWS2 score discrimination and calibration performances in the development and validation cohorts.

	Development Cohort *n* = 128	Validation Cohort *n* = 330
AUROC (95% CI)		
aPNea score model	0.86 (0.78–0.93)	0.79 (0.73–0.84)
NEWS2 score model	0.72 (0.59–0.85)	0.81 (0.75–0.87)
De Long’s test *p*-value	0.025	0.443
Brier score (95% CI)		
aPNea score model	0.108 (0.071–0.145)	0.148 (0.124–0.172)
NEWS2 score model	0.132 (0.092–0.171)	0.131 (0.109–0.152)
Wald’s test *p*-value	0.088	0.073
Spiegelhalter test z-score (*p*-value)		
aPNea score model	−0.010 (0.992)	1.267 (0.205)
NEWS2 score model	−0.045 (0.964)	−1.559 (0.119)

AUROC, area under the receiver operating characteristic curve; CI, confidence interval.

**Table 5 jcm-11-00881-t005:** Score metrics comparison based on an aPNea cut-off of 5 and a NEWS2 cut-off of 1.

Cohort	Score	AUROC (95% CI)	Sensitivity (95% CI)	Specificity (95% CI)	PPV (95% CI)	NPV (95% CI)
Validation	aPNea	0.690 (0.631–0.748)	0.934 (0.878–0.990)	0.445 (0.384–0.506)	0.335 (0.271–0.398)	0.958 (0.921–0.994)
	NEWS2	0.685 (0.624–0.746)	0.921 (0.860–0.982)	0.449 (0.388–0.510)	0.333 (0.270–0.950)	0.950 (0.911–0.989)
	*p*-value	0.837	0.200	0.906	0.917	0.675
External	aPNea	0.703 (0.597–0.810)	0.929 (0.833–1.024)	0.478 (0.360–0.596)	0.419 (0.297–0.542)	0.943 (0.866–1.020)
	NEWS2	0.539 (0.392–0.686)	0.643 (0.465–0.820)	0.435 (0.318–0.552)	0.316 (0.195–0.436)	0.750 (0.616–0.884)
	*p*-value	0.001	0.005	0.406	0.005	0.002

AUROC, area under the receiver operating characteristic. CI, confidence interval. PPV, positive predicting value. NPV, negative predicting value.

## Data Availability

Data are available from the corresponding author upon reasonable request and institutional approval.

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
