# Peer review of "Development and Validation of the Acute PNeumonia Early Assessment Score for Safely Discharging Low-Risk SARS-CoV-2-Infected Patients from the Emergency Department"

_jcm, 2022, doi:10.3390/jcm11030881_

Round 1
Reviewer 1 Report
This study is to develop and evaluate a scoring system that predicts severity using clinical parameters of COVID-19 patients who visit the emergency rooms. When the number of patients with COVID-19 is increasing rapidly, it is very important to reduce overcrowding in emergency rooms and hospital beds, so it is very important to make a good prediction model necessary to decide the patient’s hospitalization and discharge. Belows are some comments on the paper.
- I was wondering how authors assigned each point to clinical parameters in the aPNea scoring system. In the reference presented by the authors, there was no study on how to get each point in the aPNea scoring system. It is common to obtain points of each clinical parameter through relative risk or odds ratio from epidemiologic studies.
- NEWS score is a tool for predicting poor prognosis of patients, so it may not be appropriate to compare it to aPNea scoring system as an indicator of discharge of low-risk patients from ER.
- It is necessary to specify whether “Need of hospitalization for any cause” means underlying illness.
- Since primary outcome of the study was application of mechanical ventilation or death, I think it is more important to evaluate in detail how well the aPNea score can predict disease progression of COVID-19 compared to the NEWS scoring system. If ‘safety discharge’ is the main outcome of this study, I think hospitalization or ER revisit seem to be more appropriate primary outcome rather than death or ventilator application.
- Compare to NEWS score, aPNea score has the advantage of having more evaluable indicators related to respiratory symptoms and signs. However, instead of NEWS score being able to score with simple clinical parameters, aPNea score requires many tests such as lab, image tests, and 6-minute walk test.
Author Response
Authors’ reply to Reviewer 1
We thank Reviewer 1 (R1) for her/his thoughtful comments. Here is our (A) point-by-point reply to the raised concerns. Changes are highlighted in red in the revised manuscript.
R1: This study is to develop and evaluate a scoring system that predicts severity using clinical parameters of COVID-19 patients who visit the emergency rooms. When the number of patients with COVID-19 is increasing rapidly, it is very important to reduce overcrowding in emergency rooms and hospital beds, so it is very important to make a good prediction model necessary to decide the patient’s hospitalization and discharge. Below are some comments on the paper.
I was wondering how authors assigned each point to clinical parameters in the aPNea scoring system. In the reference presented by the authors, there was no study on how to get each point in the aPNea scoring system. It is common to obtain points of each clinical parameter through relative risk or odds ratio from epidemiologic studies.
A: We agree with the Reviewer that the usual way to assign points to a score is by weighting the relative risk of several prognostic factors for the outcome. However, during the first pandemic wave of SARS-CoV-2 infection in our Emergency Department, we did not have any data on the relative risk of specific factors for COVD-19 hospitalization. We built the aPNea score by looking at the parallel experience of other ED in Italy in which the clinical stronger predictive factors of disease activity were fever and respiratory failure, whereas the absence of lung involvement at imaging had a very strong negative predictive value (Bella et al. Ital J Emerg Med 2020 and Bertazzoni et al. Ital J Emerg Med 2020). We assigned a maximum of 5 points to SARS-CoV-2 positive patients that needed hospitalization for any reason independent of COVID-19 severity or patients with signs of severe respiratory failure. We graded respiratory failure severity based on the Horowitz index and by assigning 4 points to less severe respiratory dysfunction and gave 2 points for silent hypoxia diagnosed with an abnormal alveolar to arterial oxygen gradient or a positive 6-minute walk test (Tobin. Am J Resp Crit Care Med 2020 and Pandit et al. J Assoc Physicians India, 2020). The high negative predicted value of lung imaging for active COVID-19 was translated to 1 or 2 points assigned to a progressive lung disease involvement. Finally, we assigned 1 point to factors that have been associated with COVID-19 activity, such as body temperature greater than 38°C and elevated plasma D-dimer level. We have clarified this point in the method section and discussing the study limits in the revised manuscript.
R1: NEWS score is a tool for predicting poor prognosis of patients, so it may not be appropriate to compare it to aPNea scoring system as an indicator of discharge of low-risk patients from ER.
A: We agree with the Reviewer about the role of NEWS2 as prognostic score for driving hospitalization rather than discharge of patients. However, low NEWS2 score is associated with a low risk of clinical deterioration and we have previously showed that NEWS2 has the best negative predictive value for adverse outcomes when compared to other prognostic scores in septic patients (Colussi et al. BMC Emerg Med, 2021). In addition, NEWS2 was widely adopted in the Emergency Department and have showed the best prognostic value to assess COVID-19 severity in several studies (Myrstad et al. Scan J Trauma Resusc Emerg Med, 2020). Therefore, we compared our prognostic score to the best validated score for COVID-19 severity available at that time. We have specified this point in the discussion section of the revised manuscript.
R1: It is necessary to specify whether “Need of hospitalization for any cause” means underlying illness.
A: “Need of hospitalization for any cause” comprised underlying illness and other medical reasons for hospitalizing SARS-CoV-2 positive patients independent of COVID-19 severity. We have detailed this point in the result section.
R1: Since primary outcome of the study was application of mechanical ventilation or death, I think it is more important to evaluate in detail how well the aPNea score can predict disease progression of COVID-19 compared to the NEWS scoring system. If ‘safety discharge’ is the main outcome of this study, I think hospitalization or ER revisit seem to be more appropriate primary outcome rather than death or ventilator application.
A: We understand Reviewer concern about the appropriateness of our outcomes for assessing safety discharge of SARS-CoV-2 patients. Although rehospitalization or ED revisit could have been better outcomes for assessing safely discharge, 30-day death and oral intubation were outcome easily accessible in our database. This because during pandemic emergency, not all discharged patients were managed in the same hospital and some patients received home assistance. Therefore, we considered this composite outcome the most appropriated for deciding which patient to discharge safely. We have specified this point in the method section.
R1: Compared to NEWS score, aPNea score has the advantage of having more evaluable indicators related to respiratory symptoms and signs. However, instead of NEWS score being able to score with simple clinical parameters, aPNea score requires many tests such as lab, image tests, and 6-minute walk test.
A: We agree with the Reviewer that NEWS2 is simpler to calculate than our score. However, with few more detailed information easily available in all Emergency Departments, we can improve our triage process of SARS-CoV-2 infected patients. We have added this comment in the discussion section.

Reviewer 2 Report
In this study, Venturini et al. aimed to provide a new prognostic tool for COVID-19 patients admitted to emergency departments. They proposed the acute PNeumonia early assessment (aPNea) score for this purpose. This new tool provided good discrimination of low-risk patients at the end.
The manuscript is presented in a well-structured manner. The topic is novel, interesting and relevant. The design and methodology are clear. Tables are figures are adequate. Ethics and data availability statements are adequately presented. And finally, discussion is well written.
Author Response
Authors’ reply to Reviewer 2
We thank Reviewer 2 (R2) for her/his thoughtful comments. Here is our (A) point-by-point reply to concerns. Changes are highlighted in red in the revised manuscript.
R2: In this study, Venturini et al. aimed to provide a new prognostic tool for COVID-19 patients admitted to emergency departments. They proposed the acute PNeumonia early assessment (aPNea) score for this purpose. This new tool provided good discrimination of low-risk patients at the end. The manuscript is presented in a well-structured manner. The topic is novel, interesting and relevant. The design and methodology are clear. Tables are figures are adequate. Ethics and data availability statements are adequately presented. And finally, discussion is well written.
A: No concerns were risen by Reviewer 2.

Reviewer 3 Report
Dr. Venturini et al. have developed and validated a score system (aPNea) and compared the score performance to that of the NEWS2 in COVID-19 patients. They concluded that aPNea score appeared appropriate for discharging low-risk SARS-CoV-2 infected patients from ED.
The topic is interesting and the manuscript is well-organized.
Major comments:
Are all the participants presented in the ED for COVID-19 related symptoms? If not, please give the details of the presented signs and/or symptoms. How many patients have need of hospitalization for causes other than COVID-19? Please explain.
Minor comments:
Please correct the calculation error in table 1, 30-d deaths.
Author Response
Authors’ reply to Reviewer 3
We thank Reviewer 3 (R3) for her/his thoughtful comments. Here is our (A) point-by-point reply to the raised concerns. Changes are highlighted in red in the revised manuscript.
R3: Dr. Venturini et al. have developed and validated a score system (aPNea) and compared the score performance to that of the NEWS2 in COVID-19 patients. They concluded that aPNea score appeared appropriate for discharging low-risk SARS-CoV-2 infected patients from ED. The topic is interesting and the manuscript is well-organized.
Major comments: Are all the participants presented in the ED for COVID-19 related symptoms? If not, please give the details of the presented signs and/or symptoms.
A: All patients presented in ED because of mild to severe respiratory symptoms and concerns about SARS-CoV-2 infection, or for underlying illness or other medical reasons and tested positive for SARS-CoV-2 infection at hospital screening. We have specified this point in the result section of the revised manuscript.
R3: How many patients have need of hospitalization for causes other than COVID-19? Please explain.
A: Data were reported in table 3 (highlighted in red). Main reasons for hospital admission independent of COVID-19 severity were chest pain with ECG abnormalities, dehydration because of gastro-intestinal loss, hemodynamic instability, and incidental trauma. We have clarified this point in the result section of the revised manuscript.
R3: Minor comments: Please correct the calculation error in table 1, 30-d deaths.
A: We have corrected the error in Table 2, where we imagine the Reviewer saw it.

Round 2
Reviewer 1 Report
My concerns have been addressed.
Reviewer 3 Report
Thank you for the revision. I have no further comments.